# An Electrochemical Amperometric Ethylene Sensor with Solid Polymer Electrolyte Based on Ionic Liquid

**DOI:** 10.3390/s21030711

**Published:** 2021-01-21

**Authors:** Petr Kuberský, Jiří Navrátil, Tomáš Syrový, Petr Sedlák, Stanislav Nešpůrek, Aleš Hamáček

**Affiliations:** 1Research and Innovation Centre for Electrical Engineering (RICE), Faculty of Electrical Engineering, University of West Bohemia, Univerzitní 8, 306 14 Plzeň, Czech Republic; jirkanav@fel.zcu.cz (J.N.); nespurek@post.cz (S.N.); hamacek@fel.zcu.cz (A.H.); 2Department of Graphic Arts and Photophysics, Faculty of Chemical Technology, University of Pardubice, 532 10 Pardubice, Czech Republic; tomas.syrovy@upce.cz; 3Faculty of Electrical Engineering and Communications, Brno University of Technology, Technická 10, 616 00 Brno, Czech Republic; sedlakp@feec.vutbr.cz

**Keywords:** printed electrochemical sensor, solid polymer electrolyte, ionic liquid, ethylene

## Abstract

An electrochemical amperometric ethylene sensor with solid polymer electrolyte (SPE) and semi-planar three electrode topology involving a working, pseudoreference, and counter electrode is presented. The polymer electrolyte is based on the ionic liquid 1-butyl 3-methylimidazolium bis(trifluoromethylsulfonyl)imide [BMIM][NTf_2_] immobilized in a poly(vinylidene fluoride) matrix. An innovative aerosol-jet printing technique was used to deposit the gold working electrode (WE) on the solid polymer electrolyte layer to make a unique electrochemical active SPE/WE interface. The analyte, gaseous ethylene, was detected by oxidation at 800 mV vs. the platinum pseudoreference electrode. The sensor parameters such as sensitivity, response/recovery time, repeatability, hysteresis, and limits of detection and quantification were determined and their relation to the morphology and microstructure of the SPE/WE interface examined. The use of additive printing techniques for sensor preparation demonstrates the potential of polymer electrolytes with respect to the mass production of printed electrochemical gas sensors.

## 1. Introduction

Ethylene (C_2_H_4_) is a natural plant substance that controls the ripening process in many fruits and vegetables [1]. Fruits and vegetables, as a commodity group of perishable foods, are the subject of interest of the Food and Agriculture Organization of the United Nations (FAO), which focuses on food security and the reduction of food loss and waste. The FAO has published several studies [2,3] in which the loss or waste of fruits and vegetables, among other commodities, were identified along the whole supply chain from farm to consumer. However, great differences in the amounts of loss and waste among different regions across the world, the wide variety of fruit and vegetables, and the different metrics used for data analysis make it generally difficult to identify the weakest points along the whole supply chain. Nevertheless, it is not unreasonable to suggest that fruit and vegetable waste and losses can be reduced by monitoring and controlling ambient conditions such as temperature, relative humidity, and ethylene concentration during storage and transportation. In this respect, there is a strong need for a low cost and reliable ethylene sensor.

Ethylene sensors based on chemoresistive [4,5,6,7,8,9,10,11,12,13,14,15,16,17,18], optical (NDIR) [19,20,21,22], colorimetric [23,24,25], luminescence [26], fluorescence [27], piezoelectric [28,29,30], and electrochemical [31,32,33,34] principles have been developed. Each such principle has its own advantages and limitations, the comparison of which can be found in several review articles [35,36,37,38]. Here, we focus on electrochemical amperometric gas sensors because they are a well-established and versatile type of gas sensor meeting the requirements of many applications in terms of sensitivity, selectivity, and power consumption. However, these sensors face problems with liquid electrolyte leakage and/or evaporation, which usually reduces their life time. In addition to this drawback, bulky sensor housings make these sensors unsuitable for integration into modern miniature gas sensing devices.

Ionic liquids (ILs) represent promising media for the next generation of amperometric gas sensors [39,40,41] due to their unique properties such as non-flammability, sufficiently high conductivity, a large electrochemical window, structural modifiability, and excellent thermal stability. It is believed that the negligible volatility of ionic liquids can substantially improve sensor life time; in addition, their electrochemical stability allows the use of a wide range of working electrode potentials for the detection of various target gases. The conductivity of ILs is still significantly lower in comparison with conventional electrolyte solutions, but is sufficiently high for ILs to be successfully used as electrolytes in electrochemical sensors. A promising study of an electrochemical amperometric ethylene sensor was published by Zevenbergen et al. [34], in which the sensor was based on a fully-planar, three electrode topology with a gold working electrode (WE), a platinum counter electrode (CE), and a pseudoreference electrode (RE), along with a thin layer of ionic liquid, 1-butyl-3-methylimidazolium bis(trifluoromethylsulfonyl)imide [BMIM][NTf_2_] as the electrolyte. The sensor showed a swift response to ethylene exposures in the order of units of seconds, a sensitivity of 51 pA/ppm within the tested range of 0–10 ppm, and a limit of detection (LOD) of 760 ppb. However, the use of ILs requires a specific design and the fabrication of a micro-chamber for the storage of the ionic liquid electrolyte in order to prevent its leakage from the sensor housing [39,42]. Our previous study [43] demonstrated that ionic liquid can be used in polymer electrolyte, this research resulting in a fully printed, low cost electrochemical nitrogen dioxide sensor. Furthermore, the use of the solid polymer electrolyte (SPE) allows to modify the microstructure of the active electrochemical SPE/WE interface having positive impact on the sensor sensitivity [44] and the polymer electrolyte layer can be also easily patterned in order to minimize the total sensor dimension [45].

Here, we demonstrate an approach where the ionic liquid 1-butyl-3-methylimidazolium bis(trifluoromethylsulfonyl)imide [BMIM][NTf_2_] is immobilized in a polymer matrix. This allows the liquid electrolyte to be replaced by a layer of solid polymer and the sensor to be prepared with a semi-planar electrode topology, where the electrochemical active SPE/WE interface is directly exposed to the target gas. The sensor parameters are determined and their relationship to the morphology and microstructure of the SPE/WE interface is examined. The advantages and limitations of the SPE based ethylene sensor are discussed and the use of low cost additive printing techniques for the sensor fabrication is demonstrated.

## 2. Materials and Methods

### 2.1. Materials

The electrochemical sensor presented here is based on a solid polymer electrolyte consisting of three main components: 1-butyl-3-methylimidazolium bis(trifluoromethylsulfonyl)imide [BMIM][NTf_2_] ionic liquid, poly(vinylidene) fluoride (PVDF), and 1-methyl-2-pyrrolidone (NMP), all used as received from Sigma-Aldrich (Germany). The basic characteristics of the abovementioned materials, important for sensor preparation and operation, are presented below.

The electrochemical stability of the ionic liquid (IL) was determined by cyclic voltammetry (CV), because a high bias potential is required for ethylene oxidation. CV experiments were carried out using commercial screen printed electrodes (250BT, DropSens, see Figure 1) on a surface on which a droplet of IL was simply drop-cast and subsequently spread to make a thin layer. The exact thickness of the IL layer was unknown but our rough estimate based on microscopic measurement gave a value of 150 ± 50 µm. Current-voltage characteristics (cyclic voltammograms, see Appendix A) were measured by a Bipotentiostat/Galvanostat µStat 400 instrument (DropSens) in a test chamber under the following conditions: 23 °C, 40%RH, 101.325 kPa, and an analyte flow rate of 1 L/min. We achieved almost the same results as Zevenbergen et al. [34], where no electrochemical process was observed in the absence of gaseous ethylene until a working electrode potential E < 0.9 V vs. the platinum (Pt) pseudoreference electrode was applied (note that an Ag pseudoreference electrode was used in our CV experiments). At higher applied potentials, the gold working electrode was oxidized and a significant increase in measured current was observed for the applied potential E > 1.3 V due to water electrolysis. During the backward scan, a peak current of around 0.65 V indicated a reduction in the formed gold oxide. In the presence of ethylene (500 ppm), we observed a very slight increase in current within the potential range of 0.6–0.9 V, which cannot be clearly attributed to ethylene oxidation. Unfortunately, our measurement setup did not allow us to perform experiments with higher ethylene concentrations (as in Zevenbergen et al. [34]) that could validate the idea that the slight current increase was caused by ethylene oxidation.

The thermal stability was measured due to the thermal treatment of the solid polymer electrolyte (SPE) after deposition in order to achieve an SPE layer with an optimal morphology and microstructure with respect to achieving the greatest sensitivity [44]. The results of the thermogravimetric analysis (TGA) of particular components of the SPE can be found in the Appendix A. While NMP evaporates at low temperature, the ionic liquid and PVDF exhibit excellent thermal stability up to about 400 °C. All TGA analyses were carried out for a sample weight of 12 mg and at a heating rate of 5 °C/min (SDT Q600, TA Instruments).

### 2.2. Sensor Preparation

Three types of sensors with two different topologies were prepared. The first, with fully-planar topology (a reference sensor), was based on commercial screen printed electrodes (250BT, DropSens, see Figure 1) with a layer of the ionic liquid. The second, the new ethylene sensor with semi-planar topology, was based on a ceramic substrate with a Pt pseudoreference electrode (RE), a Pt counter electrode (CE), and a gold working electrode (see Figure 2). For further manufacturing details about the ceramic based platform with platinum RE and CE electrodes, see [43,46]. Two different approaches were used to prepare the sensor with semi-planar topology. The first approach aimed to fabricate both the SPE layer and the gold working electrode by means of a low cost mass production technique (screen printing, EKRA E1, in our case). The second approach focused on optimizing the structure and morphology of the SPE/WE interface in order to achieve the best sensor properties. Thus, a layer of solid polymer electrolyte was deposited by drop casting on a ceramic substrate (see [44] for further details). The SPE mixture consisted of IL, PVDF, and NMP (1:1:3, weight, respectively). For the preparation of the SPE layer by screen printing, some thinners such as dimethylformamide (DMF) and dimethyl sulfoxide (DMSO) were added to the mixture in order to tailor the viscosity of the polymer electrolyte to the screen printing process. In the case of the drop-casting method, the droplet of SPE mixture was spread on the ceramic substrate by a glass rod. Subsequently, SPE layers were thermally treated in an oven (150 °C for 10 min) or on a hot plate (120 °C for 3.5 min) for the screen printing and drop-casting methods, respectively. The above-mentioned temperatures and times were optimized experimentally because the thermal treatment conditions of the SPE layer have an impact on its adhesion properties, conductivity [47], morphology and microstructure [44], which can influence the resultant electrochemical activity of the SPE/WE interface.

The gold working electrode was deposited on the SPE layer either by screen printing (gold polymer paste C2041206P2, GWENT GROUP) or by the aerosol-jet printing (AJP) technique (gold nano-ink Au-IKTS-02, Fraunhofer IKTS). Whereas one layer (thickness 15 µm) was printed by screen printing, five layers (total thickness 2.5 µm) were printed successively by the AJP technique in order to obtain a compact gold layer. Finally, the gold working electrodes were cured in the oven at 65 °C for 30 min and 50 °C for 1 h for screen printed and aerosol-jet printed layers, respectively. Such low temperatures for gold curing were chosen in order to preserve the microstructure of the SPE layers. It should be noted that exact information about the conductivity and purity of the gold working electrode are unknown.

### 2.3. Measurement Setup

Sensor parameters were characterized by a gas test apparatus including a gas cylinder with a reference mixture of ethylene (500 ppm balanced in synthetic air), mass flow controllers (SmartTrak 100 Series, Sierra Instruments, Monterey, CA, USA), a water bubbler for gas humidification, a custom built test chamber (internal volume 90 mL), and a Bipotentiostat/Galvanostat µStat 400 instrument (DropSens, Oviedo, Spain) connected to a PC. An amperometric detection technique with a bias voltage level of 800 mV vs. Pt pseudoreference electrode was used for ethylene detection. The tested sensors were contacted via gold spring probes and covered by neither a housing nor a gas diffusion membrane. All experiments were carried out under the following conditions: 23 °C, 40%RH, 101.325 kPa, and an analyte flow rate of 1 L/min.

## 3. Results and Discussion

The sensor was exposed to several test cycles in order to determine basic sensor parameters such as sensitivity, response/recovery time, repeatability, hysteresis, limit of detection (LOD), and limit of quantification (LOQ). The first test cycle that the sensors were exposed to contained ten consecutive exposures to the same ethylene concentration (300 ppm, in our case). This test cycle provided information about the response/recovery time and repeatability of the sensor response. Figure 3a shows an example of the sensor response to a test cycle in which both exposures (to ethylene and air) alternated every 20 min. The response/recovery (T_90_/T_10_) time was defined as the time period which is necessary to achieve 90% or 10% of the full current change upon a step increase/decrease in ethylene concentration (see Figure 3b). Repeatability usually provides information about the short term stability of the sensor response. We defined it in terms of the relative standard deviation RSD (also known as the coefficient of variation) from ten consecutive exposures to the same concentration. The current value which represented each exposure was determined as follows. An average value of electric current (represented by the red square in Figure 3b) was calculated from the record for the last two minutes of the particular exposure to ethylene (grey region in Figure 3b). In this way, we obtained ten average values representing ten ethylene exposures, which served as input data for the calculation of the relative standard deviation. The second test was represented by a stepwise increase and subsequent decrease in ethylene concentration within the range 0–500 ppm (one step equaled 100 ppm). This test cycle provided data for determination of the sensitivity, LOD, and LOQ. Figure 4 shows an example of the sensor response to the test cycle where each concentration step of ethylene exposure lasted 20 min. The current value representing each concentration level was calculated as described above for the procedure determining repeatability. The red squares in Figure 4 represent average values that were calculated from the record for the last two minutes of exposure to a particular concentration level. Average values from the ascendant path were subsequently interpolated in order to obtain a calibration curve, the slope of which represented the sensor sensitivity. The sensitivity was also used to calculate the LOD and LOQ. The limit of detection was calculated as the ratio of the triple standard deviation of background current noise (at zero concentration) to sensitivity, the limit of quantification as the ratio of ten times the standard deviation of background current noise to sensitivity. Hysteresis was calculated as the difference between two current values (ΔI, see Figure 4) that were measured at the same ethylene concentration. The first value was obtained within the ascending path, the second within the descending path. The difference was subsequently expressed as a percentage of the maximum measured current (current response at the maximum ethylene concentration). The above-mentioned parameters of the tested sensors are summarized in Table 1. The first sensor was based on a fully-planar electrode topology with a layer of IL (marked as FP-250BT). The second sensor was based on a semi-planar electrode topology, where both the SPE layer and gold working electrode were prepared by screen printing (marked as SP-SCRP). The third sensor was based on a semi-planar topology where the solid polymer electrolyte layer and the gold working electrode were prepared by drop-casting and aerosol-jet printing, respectively (marked as SP-AJP). Before discussing the results in Table 1, it is necessary to remember one important fact. While the sensor with the fully planar topology (FP-250BT) had an Ag pseudoreference electrode, the new semi-planar topology contained a Pt pseudoreference electrode. This can naturally influence the actual level of the working electrode potential E and, therefore, comparisons of such sensors with different pseudoreference electrodes can be problematic or questionable. However, the fully-planar topology with the Pt pseudoreference electrode was also used by Zevenbergen et al. [34], and the cyclic voltammograms published in their study are almost identical to those published in this work. Thus, we believe that the comparison of such sensors is justifiable. We can say, therefore, that the commercial screen printed electrodes (FP-250BT) exhibited almost the same sensitivities and response times as those published by Zevenbergen et al. [34]; thus, we can use such parameters as default reference values. The higher detection limit achieved with the FP-250BT platform was primary because of the presence of greater background current noise.

### 3.1. Sensitivity

Significant progress was achieved with regard to sensor sensitivity. While the new semi-planar topology with the screen printed SPE layer and the gold working electrode (SP-SCRP) improved sensitivity only very slightly, the approach with the drop-cast SPE layer and the aerosol-jet printed working electrode (SP-AJP) resulted in more than ten times higher sensitivity. However, this result also indicated that the change in the topology itself (from fully-planar to semi-planar) did not improve the sensitivity. Thus, we focused in more detail on the SPE/WE interface, where the ethylene oxidation takes place. It should be noted that the new semi-planar topology allowed a unique insight into the morphology and microstructure of this electrochemical active interface. Figure 5a,b show both tested sensors with the semi-planar topology. First, we should mention some dimensions that were primarily determined by the different techniques of sensor preparation. The approximate thicknesses of the screen printed SPE layer and the gold WE were 10 and 15 µm, respectively. The approximate thicknesses of the drop-cast SPE layer and the aerosol-jet printed gold WE were 150 and 2.5 µm, respectively. Figure 5c,d show SEM images of the SPE/WE interface. The red ellipses in Figure 5a,b show areas that were captured by an electron microscope. These pictures indicate great differences in the morphology and microstructure of the SPE layers. While the printed layer (Figure 5c) exhibited minimum porosity with microscopic circular defects, the drop-cast SPE layer (Figure 5d) exhibited a highly porous structure. Spherical objects (so called spherulites) with diameters ranging from 10 to 50 µm were usually observed by the microscope. The morphology and microstructure of the SPE layer allowed these spherical objects to be almost completely surrounded by the gold layer, which resulted in an increase in the total electrochemically active area, and so in an increase in the sensor sensitivity. In order to exclude the influence of two different gold materials (polymer paste and nano-ink) on sensor sensitivity, we tested both paste and ink with the highly porous SPE microstructure presented in Figure 5d. Because we achieved almost the same results (regardless of the type of gold), we are convinced that the morphology and microstructure of the SPE layer had a crucial impact on the sensor sensitivity.

### 3.2. Response/Recovery Time

Interesting results were obtained by comparing the response/recovery times of the tested sensors. Whereas the most sensitive sensor (SP-AJP) had the slowest response/recovery time (6.67/6.76 min), the other mentioned sensors were more than four times faster. These results were not in line with our assumptions. We expected a slower response time for the fully-planar topology, where the working electrode was under the electrolyte layer. However, from the results presented in Table 1, we were not able to decide which topology is better in terms of response/recovery time. While one sensor with semi-planar topology showed the worst results (SP-AJP), the second sensor with semi-planar topology showed the best results (SC-SCRP). A possible explanation for this may relate to the compactness of the AJP printed gold layer. Thus, we decided to optimize the aerosol-jet printed gold layer in two ways: (i) by reducing the gold layer thickness, and (ii) by printing a gold micro-grid electrode. We observed a loss of ethylene detection capability when the total gold working electrode thickness was below 1 µm. In other words, at least three gold layers had to be printed one over the other for proper ethylene detection. It should be noted that the thickness of each layer was approximately 500 nm. This approach resulted in a decrease in the response/recovery time by four times and three times, respectively (see Table 2). However, the thinner working electrode decreased the sensor sensitivity by more than ten times. The second approach consisted of printing a grid microstructure for the working electrode (see Figure 6a) instead of reducing the layer thickness. The micro-grid pattern consisted of printed lines with a line width of 50 µm and a spacing of 100 µm. In this case, the response/recovery times were approximately two times faster. Unfortunately, this approach had a negative impact on sensor sensitivity. The sensor with the gold micro-grid working electrode showed linear behavior within the range 0–200 ppm. For higher ethylene concentrations (200–500 ppm) the current growths grew smaller and the calibration curve did not follow a linear trend (see Figure 6b). This result may be explained by the fragility of the gold micro-grid. Although five layers of the micro-grid pattern were printed one over the other, detailed microscopic observation after the experiments indicated cracks and defects in the micro-grid structure. These cracks in the printed lines could prevent charge carriers to flow from specific areas of the micro-grid, the dimensions of which are unknown; thus, parts of the electrode could be “electrically” isolated. This could subsequently result in the nonlinear behavior observed at higher ethylene concentrations. We observed similar defects in the structure of the compact gold electrode (Figure 5b). However, this compact gold layer provides many paths for electrons to flow easily around defects and, thus, such microscopic cracks cannot prevent electron flows. According to the experimental results in this section, we believe that the thickness and microstructure of the working electrode have a crucial impact on response/recovery time. However, the decrease in the response/recovery time was achieved at the expense of sensitivity and/or response linearity.

### 3.3. Repeatability

Repeatability was defined as the relative standard deviation from ten consecutive exposures to the same concentration (see Figure 3a). The worst repeatability was observed for the screen printed electrodes with fully-planar topology (FP-250BT), which could be because of the measurement of extremely low currents. However, one important fact is not clear from the numbers in Table 2. Some tested sensors showed a steadily decreasing response when they were repeatedly exposed to the same ethylene concentration. This effect grew stronger as the bias voltage applied between the RE and WE increased. Such behavior was also reported by Zevenbergen et al. [34] and the most likely explanation for it was the formation of a gold oxide layer on the surface of the working electrode, preventing further ethylene oxidation. On the one hand, a higher bias voltage resulted in a higher sensor response. On the other hand, a higher bias voltage resulted probably in the greater formation of gold oxide and thus poor repeatability. Thus, the bias voltage level had to be chosen carefully in order to balance both requirements.

### 3.4. Limit of Detection and Quantification

Both detection and quantification limits are tightly bound with sensor sensitivity and background current noise. Therefore, obviously, when the background noise level does not change significantly, the most sensitive sensor has the lowest LOD and LOQ. Both the LOD and LOQ (0.8 and 2.6 ppm, respectively) were verified experimentally (see Figure 7), with the calibration curve measured in more detail from 0 to 100 ppm. The inset in Figure 7 shows the current-time response of the sensor to stepwise increases from 0 to 10 ppm, where one step equaled 2 ppm. Comparison of the limits of detection in Table 2 provides further information. While the sensor with semi-planar topology (SP-AJP) exhibited approximately twenty-two times higher sensitivity than the sensor with fully-planar topology presented by Zevenbergen et al. [34], the calculated LOD was surprisingly slightly higher. This means that the sensor with semi-planar topology (SP-AJP) generated much higher background current noise. In spite of the fact that noise is usually considered an unwanted component of the sensor signal, it can provide valuable information about physical and/or chemical stochastic processes taking place at/on the electrochemically active interface/surface. This situation was exploited in our previous studies [48,49,50], where the analysis of current fluctuations enabled additional information about physicochemical processes at the SPE/WE interface of the electrochemical sensor to be obtained. Such analysis, however, is beyond the scope of this paper, which is aimed at studying the morphology and microstructure of the SPE/WE interface and the influence of this interface on sensor parameters. Nevertheless, we believe that this work may provide a solid basis for further fluctuation-based analysis, which may open up a pathway towards further improvement of the detection capability of such amperometric gas sensors.

### 3.5. Operating Conditions and Limitations of SPE Based Amperometric Ethylene Sensor

In the following section, some important facts (that are not evident from the previous text) are mentioned and discussed.

All experiments were carried out under constant and defined conditions, including temperature, relative humidity, pressure, and analyte flow rate, in order to eliminate parasitic effects of these quantities on sensor response. We experimentally tested the influence of relative humidity (RH) at zero ethylene concentration, when an increase from 30%RH to 50%RH resulted in a current increase of 15 nA, which roughly represents an equivalent increase in ethylene concentration of 13 ppm. This effect should be compensated when precise ethylene determination is required.

A high bias voltage used for ethylene oxidation (0.8 V vs. Pt pseudoreference electrode) has two impractical consequences. The first, invisible in our figures, resides in the so called “warm-up” time of the sensor. It is usually defined as the time period which is necessary for the stabilization of the background current flowing through the sensor at zero concentration of target gas after the power supply is switched on. It is not constant but increases with the applied bias voltage. In our study, a bias voltage of 0.8 V meant a warm-up time of between 30 and 45 min. The second consequence resides in possible cross-interference with other gases, the oxidation potential of which is lower than 0.8 V. Exploring these effects was beyond the practical scope of this research, but could be the subject of further work.

### 3.6. Advantages of SPE Based Amperometric Sensors—Towards a Printed Sensor on the Flexible Substrate

With manufacturing in mind, we also prepared the presented ethylene sensor with semi-planar electrode topology on a flexible substrate, substituting the original ceramic substrate with a sheet of Kapton substrate. This approach, where both electrodes and electrolyte can be processed using low-cost additive printing techniques, highlights the potential of polymer electrolytes with respect to the mass production of fully printed electrochemical gas sensors. The platinum counter and pseudoreference electrode were either printed by AJP technique (nanoparticle platinum ink Pt-LT-20, Fraunhofer IKTS, curing temperature 200 °C for 30 min) or prepared by “lift-off” technology (see Figure 8a,b). Subsequently, the SPE layers were deposited either by drop-casting or by screen printing (Figure 8c,d). The SPE compositions and thermal treatment conditions are described in Section 2.2. The gold working electrodes (WE) were printed either by AJP or screen printing (Figure 8c,d). The curing conditions for gold working electrodes are also described in Section 2.2. The sensors on flexible substrates were exposed to the test cycles described in Section 3 and resulting sensor parameters are summarized in Table 3. We can reasonably say that slight differences in the presented parameters between the flexible (Table 3) and rigid (Table 1; Table 2) platform can be attributed to the sample-to-sample variation during the preparation procedure. Figure 9 shows sensor response to repeated stepwise increase in ethylene concertation within the range 0–500 ppm (one step equals 100 ppm). This result demonstrates swift and reproducible sensor response within the tested range. While a slight drift of the background current at zero ethylene concentration can be attributed to the long “warm-up” time described in Section 3.5, different levels of the background current for both sensors is probably caused by different composition and properties of the polymer electrolyte layer.

## 4. Conclusions

An electrochemical amperometric ethylene sensor with a new form of semi-planar electrode topology is presented. This design brings several advantages, such as the absence of liquid electrolyte, the better sensitivity of SPE based ethylene sensors, and the opportunity to study the microstructure of the gold working electrode and the SPE/WE interface in more detail. While the microstructure of the polymer electrolyte had a substantial impact on sensor sensitivity, the morphology of the gold working electrode strongly influenced the response/recovery time. From the technological point of view, the use of the polymer electrolyte allowed the sensor to be prepared by means of a low-cost, mass production technique, which highlights the potential of SPE based electrochemical gas sensors. However, in spite of the promising values achieved for the sensor parameters, the performance of the presented sensor is still relatively low with respect to being commercially useful in industry, where many unwanted effects must be taken into account. Further research—either in the field of ionic liquids to achieve more effective ethylene oxidation, or on the development of detection techniques that can obtain additional valuable information—will be necessary in order to create a commercially competitive, SPE based electrochemical ethylene sensor.

## Figures and Tables

**Figure 1 sensors-21-00711-f001:**
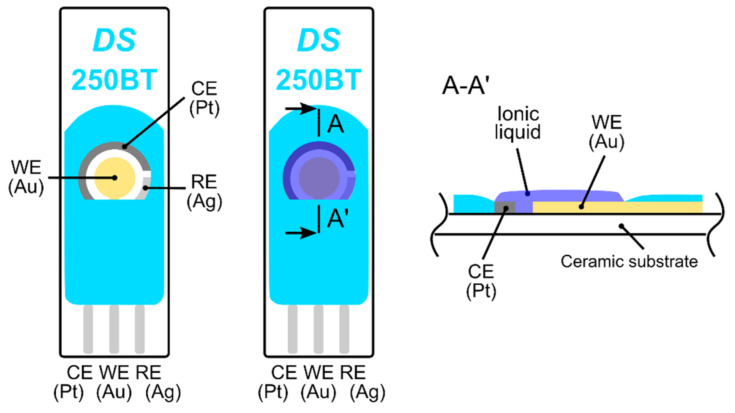
Fully-planar screen printed electrodes (250BT, DropSens); from the left: top view of the electrode topology, top view and cross section of the electrode topology with the ionic liquid layer.

**Figure 2 sensors-21-00711-f002:**
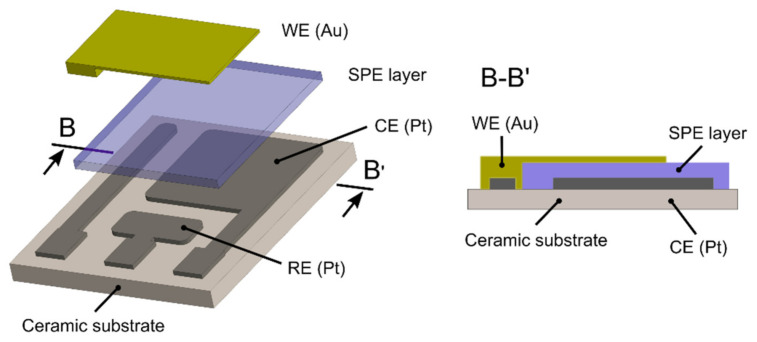
Semi-planar topology of the new ethylene sensor.

**Figure 3 sensors-21-00711-f003:**
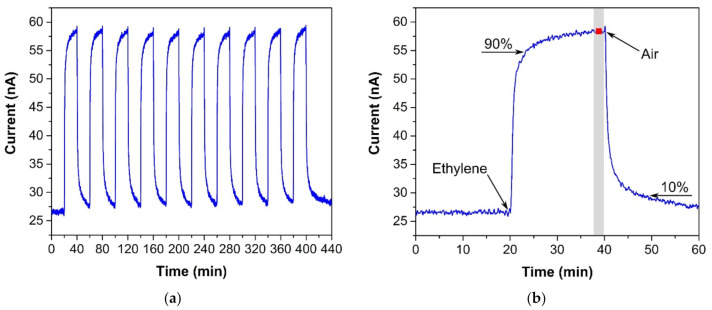
(**a**) Sensor response to ten consecutive ethylene exposures (300 ppm), (**b**) Sensor response to one ethylene exposure with details for determining the response/recovery time and repeatability. Conditions: 22 °C, 40%RH, 101.325 kPa, 1 L/min.

**Figure 4 sensors-21-00711-f004:**
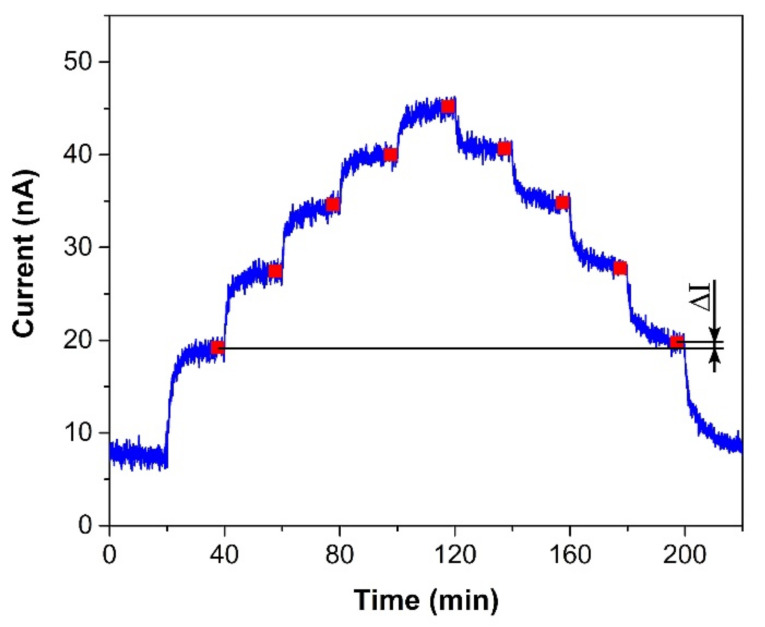
Sensor response to a stepwise increase/decrease in ethylene concentration within the range 0–500 ppm (one step equals 100 ppm). Conditions: 22 °C, 40%RH, 101.325 kPa, 1 L/min.

**Figure 5 sensors-21-00711-f005:**
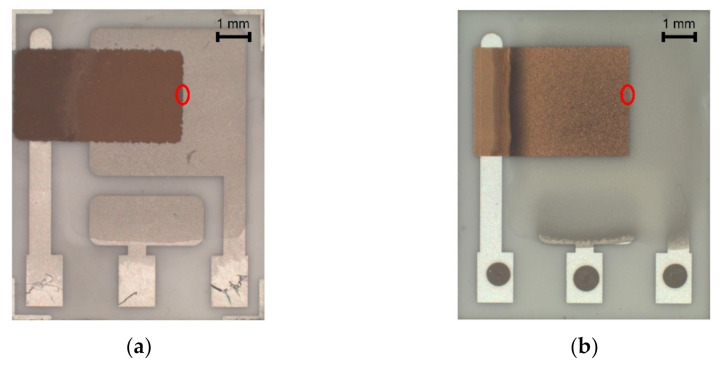
(**a**) Sensor with semi-planar topology and screen printed solid polymer electrolyte (SPE) layer and working electrode (WE) (SP-SCRP), (**b**) Sensor with semi-planar topology and drop-cast SPE layer and aerosol-jet printed WE (SP-AJP), (**c**) SEM image of the SPE/WE interface of the SP-SCRP sensor, (**d**) SEM image of the SPE/WE interface of the SP-AJP sensor.

**Figure 6 sensors-21-00711-f006:**
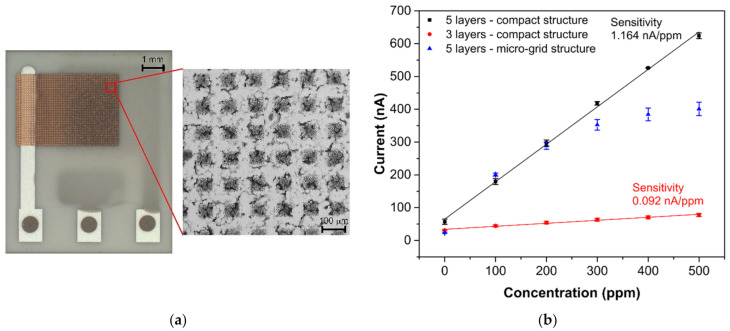
(**a**) Sensor with semi-planar topology and gold “micro-grid” electrode, (**b**) Calibration curves of sensors with semi-planar electrode topology and different gold working electrode structures (each point represents a mean value of 3 repeated exposures with corresponding 95% confidence interval). Conditions: 22 °C, 40%RH, 101.325 kPa, 1 L/min.

**Figure 7 sensors-21-00711-f007:**
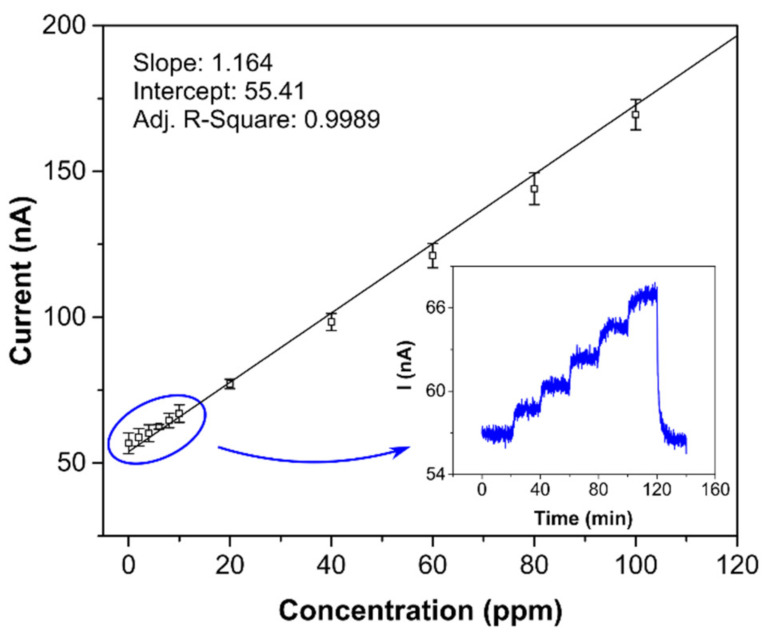
Calibration curve for the sensor with semi-planar topology and aerosol-jet printed compact Au electrode (5 layers); each point represents a mean value of 3 repeated exposures with corresponding 95% confidence interval; inset: current-time record within the range 0–10 ppm in steps of 2 ppm. Conditions: 22 °C, 40%RH, 101.325 kPa, 1 L/min.

**Figure 8 sensors-21-00711-f008:**
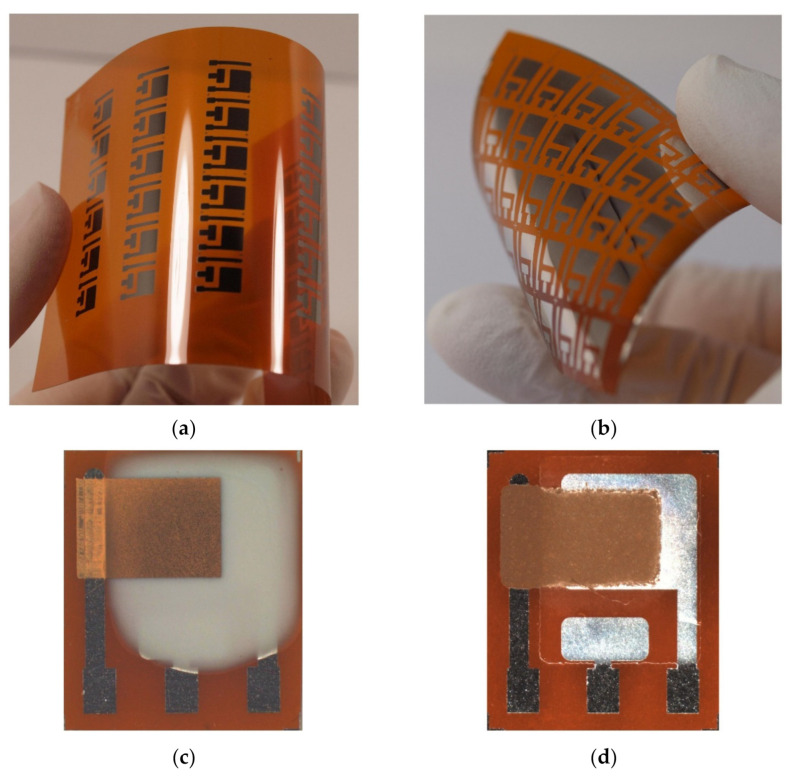
The amperometric ethylene sensor with the semi-planar topology on the flexible Kapton substrate: (**a**) Platinum pattern of the pseudoreference (RE) and counter (CE) electrode printed by the AJP technique, (**b**) Platinum pattern of the RE and CE fabricated by “lift-off” technology, (**c**) The ethylene sensor with the drop-cast SPE layer and AJP-printed gold WE, (**d**) The ethylene sensor with the screen printed SPE layer and gold WE.

**Figure 9 sensors-21-00711-f009:**
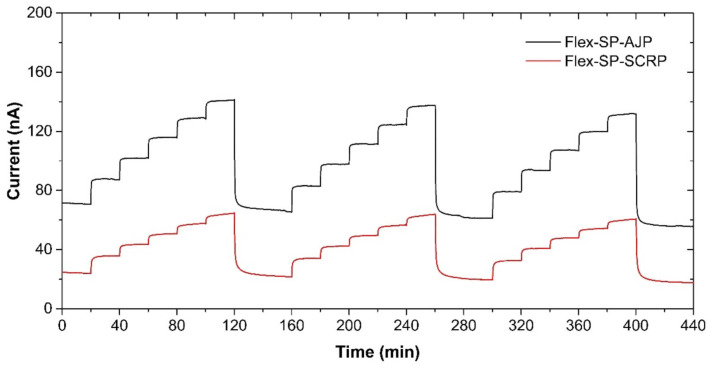
Sensor response of the semi-planar topology on the flexible Kapton substrate to the stepwise increase in ethylene concentration within the range 0–500 ppm in steps of 100 ppm. Conditions: 22 °C, 40%RH, 101.325 kPa, 1 L/min.

**Table 1 sensors-21-00711-t001:** Sensor parameters.

Parameter ^1^	Zevenb. et al. [34]	FP-250BT	SP-SCRP	SP-AJP
Sensitivity (nA/ppm)	0.051	0.064	0.074	1.164
Response time (min)	1.5	1.48	0.98	6.67
Recovery time (min)	–	1.4	1.18	6.76
Repeatability (%)	–	27	2	4
Hysteresis (%)	–	6.7	3.3	4.4
LOD (ppm)	0.77	39	26	0.8
LOQ (ppm)	–	131	85	2.6

^1^ All parameters obtained at the bias voltage of 800 mV vs. Ag or Pt pseudoreference electrode at the following conditions: 22 °C, 40%RH, 101.325 kPa, 1 L/min.

**Table 2 sensors-21-00711-t002:** Optimization of aerosol-jet printing (AJP) printed gold working electrode.

Parameter ^1^	5 Gold Layers (Compact Structure)	3 Gold Layers (Compact Structure)	5 Gold Layers (Micro-Grid Structure)
Sensitivity (nA/ppm)	1.164	0.092	–
Response time (min)	6.67	1.65	3.82
Recovery time (min)	6.76	2.26	3.76

^1^ All parameters obtained at the bias voltage of 800 mV vs. Pt pseudoreference electrode at the following conditions: 22 °C, 40%RH, 101.325 kPa, 1 L/min.

**Table 3 sensors-21-00711-t003:** Parameters of ethylene sensors on the flexible substrate.

Parameter ^1^	Flex-SP-AJP ^2^	Flex-SP-SCRP
Sensitivity (nA/ppm)	0.14	0.078
Response time (min)	2.33	0.67
Recovery time (min)	3.88	0.83
Hysteresis (%)	10	10
LOD (ppm)	0.8	0.9
LOQ (ppm)	2.6	2.9

^1^ All parameters obtained at a bias voltage of 800 mV vs. Ag or Pt pseudoreference electrode under the following conditions: 22 °C, 40%RH, 101.325 kPa, 1 L/min. ^2^ Three gold layers were printed successively (one over the other).

## Data Availability

The data presented in this study are available on request from the corresponding author.

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
