# Peer review of "An Electrochemical Amperometric Ethylene Sensor with Solid Polymer Electrolyte Based on Ionic Liquid"

_sensors, 2021, doi:10.3390/s21030711_

Round 1
Reviewer 1 Report
The manuscript entitled An electrochemical amperometric ethylene sensor with solid polymer electrolyte based on ionic liquid submitted by the group of Authors represents a new form of semi-planar electrode topology for electrochemical amperometric ethylene detection.
The introduction part should be updated with some new references.
In measurements and data presentations, the Authors did not include the number of repetitions and RSD vales.
What is the lifetime of proposed sensors?
Descriptions and images in Figure 2 should be separated to be more clear.
My compliments to the Authors for giving such and realistic evaluation of proposed sensors in the terms of performances and applications compared to current state of the art in the field. The research is interesting and has a potential. For this reason I would encourage the Authors to continue to develop proposed sensor.
Reviewer 2 Report
This work described an amperometric ethylene sensor based on the IL modified commercial sensor. I think this work is worth to be accepted for publication after a minor revision. Please find attached specific comments.
- The results in Figure 1 should put into supplementary information without include in the main content.
- Error bars should include in Figure 6B and Figure 7.
- As shown in Figure 7, the sensitivity is 1.164 in regards to the 1 ppm, while the background has 55.41 nA. Can you compare your sensitivity to previous reports?
- How about the anti-interference property?
- Can you described the RSD of using a batch of sensors?
